



# 19th–20th century semi-quantitative surface ozone along
# subtropical Europe to tropical Africa Atlantic coasts

Juan A. Añel[1,2], Juan-Carlos Antuña-Marrero[1,3], Antonio Cid Samamed[4], Celia Pérez-Souto[1], Laura de
la Torre[1], Maria Antonia Valente[5], Yuri Brugnara[6], Alfonso Saiz-López[2], Luis Gimeno[1]
[1]EPhysLab, CIM-UVigo, Universidade de Vigo, Ourense, 32004, Spain
[2]Department of Atmospheric Chemistry and Climate, Institute of Physical Chemistry Blas Cabrera,
CSIC, Madrid, 28006, Spain
[3]Grupo de Óptica Atmosférica, Facultad de Ciencias, Universidad de Valladolid, Valladolid, 47011,
Spain
[4]Physical Chemistry Department, Faculty of Sciences, Universidade de Vigo, Campus de As Lagoas
S/N, 32004 Ourense, Spain
[5]Instituto Dom Luiz, Faculty of Sciences, Universidade de Lisboa, Lisboa, 1749-016, Portugal
[6]Empa, Laboratory for Air Pollution/Environmental Technology, Dübendorf, Switzerland
*Correspondence to*: Juan A. Añel (j.anhel@uvigo.gal)

**Abstract.** Tropospheric ozone ($O_3$) plays a key role in the climate system. Studying pre-industrial
tropospheric $O_3$ implies two important challenges: *i*) the lack of observational records prior to the late
19th century, which hampers understanding long-term climate trends, given $O_3$ crucial role, *ii*) and the
uncertainties on their quantitative values in a non-polluted atmosphere across the planet. The
ozonoscope was the first instrument used to measure ozone. It offers semi-quantitative estimates of
surface $O_3$ when no other measurements were available. Despite their potential value, the digitisation,
curation and publication of ozonoscope data remains largely unexplored. In this work, we initiate an
effort to rescue surface $O_3$ ozonoscope records with a new data collection. We include data from 23
observatories covering Portugal and the African Atlantic regions, providing a latitudinal span from the
extratropics in the northern hemisphere to the tropics in the southern hemisphere. This record represents
the most extended ozonoscope data series to date, spanning 50 years of daily data and 58 years of
monthly data, from 1855 to 1913.

## 1. Background & Summary
Tropospheric ozone ($O_3$) records for our planet before the end of the 19th century are rare and sparse. It
is not surprising given that $O_3$ was discovered by Schönbein in 1839 (Schönbein, 1840a, b) and it got
little attention during the decades following its discovery. After 1860s, measuring it became common at
meteorological stations. However, $O_3$ is a transcendental chemical for the understanding and study of
the atmosphere. Tropospheric $O_3$ is a greenhouse gas and, at elevated concentrations, a pollutant
harmful to human health, also affecting crops and ecosystems productivity (U.S. EPA, 2020). However,
the study of tropospheric $O_3$ faces two important challenges: *i*) the lack of observational records prior
to the late 19th century, which hampers understanding long-term climate trends, *ii*) and the uncertainties
on the quantitative values in a non-polluted atmosphere.
The first phase of the Tropospheric Ozone Assessment Report (TOAR) project [(Schultz et al., 2017; Tarasick et al., 2019)]
developed a web-accessible database of surface $O_3$ observations, consisting in two main periods. The
modern period, beginning around 1975 and spanning to the present, defined by widespread availability
of sensitive UV photometers for surface $O_3$ measurements, and the historical period, covering 1877–
1975, defined by the use of other techniques and the lack of UV photometers. The records available for
the period previous to 1975 were evaluated using a set of four criteria to minimize uncertainties and
biases between the measurement techniques available at that times, and the contemporary UV
absorption standard. Those criteria are: the relationship of the measurement technique to the modern
UV absorption standard, the absence of interfering pollutants, the representativeness of the well-mixed
boundary layer, and expert judgement of their trustworthiness. The earliest surface $O_3$ measurements,
corresponding to the 19th century, extending until the early 20th century using the test-paper method,
also called "ozonoscope", were among the ones disregarded (Tarasick et al., 2019).
Considering the scientific questions motivating the TOAR, associated with the global distribution and
trends of surface $O_3$ pollution [(Gaudel et al., 2018)], the decision not to include the 19th century semi-quantitative
$O_3$ measurements in the TOAR database is grounded. Yet, there are other scientific questions related to
global distribution and surface $O_3$ pollution during the pre-industrial era, such as the atmospheric





concentration in non-polluted areas and evaluation of the assumed $O_3$ concentrations, study of local
sources of $O_3$, and better understanding of the role of such levels on the radiative balance. Answering
such questions could benefit of using together the quantitative surface $O_3$ observations with semi-
quantitative $O_3$ observations from ozonoscopes. Although the measurements with the ozonoscope were
vulnerable to the influence of the humidity and oxidants in the air, those semi-quantitative $O_3$
observations will enable us to study semi-quantitatively climate variables under very low (or no
exposure at all) anthropogenic activity when no other measurement was available (Bojkov, 1986), a gap
in our knowledge of surface $O_3$ in the 19[th] century.
Efforts to recover some of those $O_3$ measurements have been performed in the past (Bojkov, 1986;
Linvill et al., 1980; Anfossi et al., 1991; Sandroni et al., 1992; Sandroni and Anfossi, 1994; Marenco et
al., 1994; Cartalis and Varotsos, 1994; Nolle et al., 2005); however, only a single sample of those
surface $O_3$ datasets that we know of was digitized, published in a public data repository, and it was
done during data recovery efforts focused on something other than ozone or atmospheric composition
(Vaquero et al., 2022).
Here we introduce the rescued surface $O_3$ ozonoscope records covering Portugal and the African
Atlantic oceanic sector from 23 observatories in four countries. The $O_3$ semi-quantitative observations
provide a latitudinal coverage from extratropics in the northern hemisphere to tropics in the southern
hemisphere. The observations were conducted following a standardized procedure with the same type
of test-paper (Schönbein, 1850; Bérigny, 1858). The series of daily and monthly means of surface $O_3$
and humidity, and their corresponding metadata, have been digitized from the original documentary
sources. They are representative of very different regions of the planet, such as tropics, oceans and
coastal areas. One of them, from the Infante D. Luiz observatory, located in Lisbon, Portugal, provides
almost fifty years of continuous daily data in the period 1863 to 1913 and nearly fifty-eight years of
monthly means from 1855 to 1913, becoming the most extended and earlier surface $O_3$ ozonoscope data
series known to date. Before this work, the longest and earliest reported series was the thirty-one years
Montsouris observatory $O_3$ ozonoscope data series, which began in 1876 (Bojkov, 1986). Additionally,
another ten of the daily records here recovered cover between twelve and seventeen years of data, while
four monthly mean records extend between thirty-three and forty-two years. The difficulty to find
records of meteorological variables covering oceanic regions for the preindustrial era, and in particular
surface $O_3$ datasets, makes this contribution one of the most relevant features of the datasets recovered
here, as it contains six datasets from islands in the East Atlantic (two at the Azores Island, one in the
Madeira Island, two in Cape Verde and another in Saint Thomas & Prince). The data series have been
tested for breakpoints and inhomogeneities, finding few of them; unfortunately, we have discovered
scarce metadata that let us to provide context for the existing breakpoints; however, in several cases a
change in the location of the observatory or instruments seems a plausible explanation.
In the next section, the Schönbein test paper method and its further improvement by Berigny are briefly
described, followed by the description of the data sources. Then, in Data Records, we describe the main
features of the recovered datasets, both for the daily and monthly means of the Infante D. Luiz
observatory and the other twenty-two observatories. The Technical Validation explains the
homogeneity tests applied.

## 2. Methods
### 2.1. The test-paper method

The test-paper measurement method was based on the color change of an indicator test paper. The strip
of blotting paper was coated with starched potassium iodide and then exposed to air between eight and
twenty-four hours, protected from solar radiation and rain. After the exposure, the strip was moistened,
developing a bluish color associated with the formation of a complex between starch and iodide,
produced by the reaction between $O_3$ and iodide. The coloration depends on the $O_3$ concentration.
Finally, the observed color was compared with a standard chromatic scale, graduated by Schönbein
from 1 to 11, proportional to the $O_3$ content in the air (Schönbein, 1850; Ramirez-Gonzalez et al.,
110 2020).

The method was criticized after it began to be used because the paper strip changes its color depending
on the extent of the iodide reaction with ozone, but also with humidity and other atmospheric oxidants
(Houzeau, 1857; Fox, 1873). When air reaches its water vapor saturation it causes the pre-dried paper
to humidify increasing the rate of $O_3$ absorption (Kley et al., 1988; Volz and Kley, 1988). Those are the
reasons for a non-linear correlation between the color changes and the ozone concentration.
Bérigny introduced the Schönbein's method in France in 1856 (Bérigny, 1856a, b, 1857). He also
improved the method, defining the operating procedure, presenting a more precise chromatic scale
graduated from 0 to 21 (Bérigny, 1858), and selecting the best quality of impregnated paper, the



Berzelius paper manufactured by Jame, a chemist at Sedan (France) (Marenco et al., 1994). This scale
was often referred to in logbooks containing measurements as "Jame de Sedan".
More than one and half century after the test-paper method was introduced, numerous research has been
conducted to understand the physical-chemical processes involved in the method and to deal with the
associated interference problems (Marenco et al., 1994). Those studies for example estimated that the
$O_3$ levels for 1880 to 1900 were approximately 10 ppb in the Great Lakes area of North America, with
an annual cycle maximum in April-June, and the minimum in October-November (Bojkov, 1986).
Another study using observations from Montevideo, Uruguay (1883-1885) and Cordoba, Argentina
(1886-1892) also showed $O_3$ levels of the order of 5 – 10 ppb (Sandroni et al., 1992). It is beyond the
scope of this article to discuss all the reported studies; we refer the readers to the review conducted by
Marenco et al. (1994). Among the cited interferences present in the Schönbein method the one
originated by the humidity has been the focus of multiple studies (Fox, 1873; Houzeau, 1857; Linvill et
al., 1980; Bojkov, 1986; Marenco et al., 1994; Ramirez-Gonzalez et al., 2020). For this reason, the
dependence of the ozonosonde values on humidity, here we also compile daily humidity values for the
same days the $O_3$ observations were conducted at each site.
All the twenty-three sites, whose $O_3$ observations are reported here, followed a standardized procedure
and used the same test-paper. The $O_3$ observations were conducted following the Schönbein method
with the improvements introduced by Bérigny and using the Jame (de Sedan) paper. However, the
observations in the Bérigny scale were converted to the decimal Schönbein scale, the one used for its
processing and reporting (Fradesso da Silveira, 1865). For most of the observatories two strips of paper
were exposed in the period of twenty four hours, reporting measurements each twelve hours. However,
at some of the observatories only one strip of paper was exposed with the measurement lasting for
twenty four hours, resulting in one daily observation. Further details on the exposure at each
observatory are described below.

## 2.2. Data source: Annaes do Observatorio do Infante D. Luiz

The available information about the meteorological and magnetic observations conducted at the
observatory Infante D. Luiz and its twenty-two associated observatories consist of the climatological
tables reporting daily, monthly and seasonal means of the observed variables. They included $O_3$,
published for the first time in 1863, beginning the series of "Annaes do Observatorio do Infante D.
Luiz" (hereinafter AOIDL) reports (Brito Capello, 1863). $O_3$ observations were reported in that 1[st]
volume only for the Infante D. Luiz observatory, consisting of the monthly and seasonal means of the
diurnal, nightly and daily mean, from December 1855 to November 1863. The 2[nd] volume, the
following year, began to include the daily diurnal and nightly $O_3$ observations at Infante D. Luiz from
December 1863 to November 1864 (Fradesso da Silveira, 1864). The subsequent volumes of the
AOIDL, continued including the daily diurnal and nightly $O_3$ observations at Infante D. Luiz until
November 1913 [(De Almeida Lima, 1913)]. The reports from 1914 for the Infante D. Luiz observatory still
contained the diurnal and nightly daily observations for all the variables and the columns for $O_3$ were
filled with 0,0 or " – " (De Almeida Lima, 1914). No information has been found for the end of the $O_3$
observations in 1913 at this observatory.
In addition to the Infante D. Luiz $O_3$ diurnal observations, the 2[nd] volume published in 1864 included
decadal, monthly, and annual $O_3$ means from other observatories. The following volumes of the AOIDL
continued reporting the monthly and seasonal means of the diurnal, nightly and daily mean for the
Infante D. Luiz observatory (De Almeida Lima, 1913). Again, no information has been found for the
interruption of the $O_3$ observations.
The reports of the decadal and monthly $O_3$ means at the associated twenty-two observatories continued
after 1864, until 1905 (De Lina Vidal, 1905). Daily $O_3$ observations from some of those observatories
conducted in December 1872 began to be reported intermittently, at least in the records we have already
found, in the volume 11, corresponding to 1873 (Fradesso da Silveira, 1873). In the available AOIDL
reports we found daily observations for some of the observatories ending in 1887, although the decadal
and monthly means continued being reported. No reason has been found for the interruption of the
reports; we speculate that the cost of publishing them could be a cause for it, a common reason in many
cases. The rescued metadata comes mainly from the several sections (Introduction, Advertency, etc.)
included randomly in the AOIDL (Brito Capello, 1863).
An advantage of this work is that all the ozonoscopes of the different observatories were calibrated in
the Infante D. Luiz observatory. Figure 1 shows the geographical distribution of the observatories,and
Table 1 lists them, together with their geographical coordinates and their altitude, in decreasing latitude
order to facilitate their identification on Figure 1. Figure 2 provides an example of the tables in the
AOIDL containing the recovered $O_3$ data.






### 3. Data Records
#### 3.1. Daily and monthly mean $O_3$ series from Infante D. Luiz observatory

The $O_3$ observations at the Infante D. Luiz observatory began in January 1855, together with a set of
meteorological observations (Silvestre, 1881), and continued uninterrupted until October 13th, 1913 (De
Almeida Lima, 1914). Between 1853 and September 1863 the station was in a building, with the
coordinates 38° 43' 13" N, 9° 8' 20" W. The station was moved to a different building in October 1863,
where it has remained until January 1st 1941, located at 38° 42' 59" N, 9° 8' 56" W (De Almeida Lima,
1918; Mendes Víctor, 2001) (at this date, the meteorological instruments were moved from the top of
the main building to a new meteorological park next to this building, officially maintaining the same
geographical coordinates). However, the measurement of its new location was conducted in 1879 and
reported by first time in 1881 (Capello, 1881). No further changes were reported, at least until the end
of the data series considered here. Therefore, we have assumed the last reported coordinates for the
measurements conducted after October 1863.
Table 2 shows the yearly coverage of the rescued daily series, consisting of almost fifty years of data
from 1863 to 1913. The daily observations from 1855 to 1862 were not included in the 1st AOIDL
volume [Brito Capello, 1863]. However, the monthly means of observations were included, as shown in Table
3. That is the reason for the difference between the number of years of data rescued for the daily and
monthly means for the Infante D. Luiz observatory. Both series are by far the longest reported in the
literature. They are also among the earliest.
#### 3.2. Daily and monthly mean $O_3$ series from the other 22 observatories

For four of the stations reported in the AOIDL reports, Alcanhoes, Beja, Faro and Sao Fiel, from 1863
to 1914, we found only monthly mean values. This is why nineteen stations are listed in Table 2 (daily
data) and twenty-three in Table 3 (monthly data). Also, Table 2 shows that although two daily $O_3$
observations were conducted at least at nine observatories, only the two daily observations at the
Infante D. Luiz observatory were included in the cited AOIDL reports.
In the existing literature, daily and monthly mean $O_3$ observations at Campo Maior had already been
recovered (Vaquero et al., 2022), and stored in the PANGAEA open access dataset repository (Vaquero
et al., 2021). The monthly means series here reported match the one they reported. However, the daily
$O_3$ observations (Brito Capello, 1877) did not contain the observations for the period 1863 to 1872,
which we include in the recovered observations reported here. Also daily mean $O_3$ observations at the
Porto observatory from 1861 to 1897 were reported (without making the dataset available) (Alvim-
Ferraz et al., 2006). Table 2 shows we were only able to find and recover daily mean $O_3$ observations
from Porto between 1872 and 1887; we were more successful regarding the monthly mean $O_3$
observations, shown in Table 2, recovering the period from 1862 to 1877 and the years 1897, 1900 and
1901. Daily mean $O_3$ observations from the Luanda observatory between 1890 and 1895 were used
(again without making the dataset available) (Pavelin et al., 1999). On top of them, here we have been
able to recover eight additional years of daily and monthly mean $O_3$ observations from the Luanda
observatory, from 1880 to 1887. Neither the Porto's nor the Luanda's $O_3$ datasets described here had
been reported or published in data repositories.
### 4. Technical Validation
#### 4.1. Recovered datasets quality control

Each variable from the datasets was checked, assuring that they were in the range of its respective
physically plausible magnitudes, the so-called limit test (Vaquero et al., 2022), and therefore the
consistency of its recorded values.
The homogeneity of the recovered data was tested using the software Climatol 4.0.0 (Guijarro, 2023),
which is based on the Standard Normal Homogeneity Test (SNHT) (Alexandersson, 1986). Climatol
reconstructs each time series using the data from the neighboring stations and uses the reconstructed
series as a reference to check homogeneity. Among the parameters that are set by the user, two are
particularly relevant: the distance at which the weight of the reference stations is halved, and the
threshold of the SNHT statistic above which an inhomogeneity is considered significant. The former
parameter was set at 1000 km, the latter at 25 (the default value) for $O_3$ and 15 for RH. The
measurements taken in Lisbon before 1863 and after 1905 were not checked because of the absence of
reference stations. The full results of the homogeneity test are provided as Supplementary Information
to this paper.



**4.2. Breakpoints**


Forty-seven breakpoints were identified in the $O_3$ series and forty-six in the RH series (in 17 stations).
Six breakpoints of the $O_3$ series coincide with those of the RH series. Additionally, in four of these
stations the $O_3$ breakpoints happen later in the series than those of RH, within an interval of six months
with respect to the RH observations (see Tables 2 and 3).
It is noteworthy that for Angra de Heroismo (on 1891-10-01) and Ponta Delgada (on 1867-12-01), the
$O_3$ and RH breakpoints coincide in the same month.
The Montecorvo series has two simultaneous breakpoints for $O_3$ and RH, with a difference of 8 years.
But in the first, the RH breakpoint (on 1879-10-01) precedes that of $O_3$. The other $O_3$ breakpoint (on
1887-09-01) happens after the RH breakpoint (on 1887-06-01).
In three of these cases, the coincidence of the interval between $O_3$ and RH is reduced to one or two
months: Angra de Heroismo, Ponta Delgada and Villa Fernando, which could point out to a similar
condition (e.g. moving the observatory or instruments) between both changes.
Despite the lack of metadata supporting it, simultaneous breakpoints in both $O_3$ and RH series could
point out a change in the location of the station, as two independent instruments and data series
simultaneously suffer alterations. We searched in the AOIDL for information to identify the possible
causes of the breakpoints, but we only found information about the Loanda observatory, which is
speculated was moved in 1881 (Raposo, 2017), when its RH data shows two breakpoints. For the
remainder we did not found anything. There was a slight change in the geographical coordinates of the
Infante D. Luiz observatory in 1879, as already described above, but it is unlikely it was a change in the
site location. The only changes in the observations were found at São Tome: the initial twenty-four
hour strip exposure between 3 PM of consecutive days were reported from March 1873 to January 1882
and in the February report the same year observations change to twelve hour strip exposures from 9
AM to 9 PM and 9 PM until 9 AM of the following day, depicted on Table 2. At this site two
breakpoints in the $O_3$ series were detected in October 1874 and January 1886. No RH breakpoints were
reported.

**5. Data Availability**
The surface $O_3$ semi-quantitative monthly (Añel et al., 2024b) and daily (Añel et al., 2024a) datasets
recovered and reported here have been deposited at PANGAEA, and they are available at
https://doi.org/10.1594/PANGAEA.969241 and https://doi.org/10.1594/PANGAEA.969259
respectively.
**6. Code Availability**
The Climatol 4.0.0 (Guijarro, 2023) software (DOI: 10.5281/zenodo.12786007) was used for the
homogeneity test. For computational reproducibility (Añel, 2011, 2017) it is distributed as free software
under the GPLv3 license and stored in a permanent Zenodo.org repository
(https://zenodo.org/records/12786077).

**7. Conclusions**
We have recovered semi-quantitative surface $O_3$ ozonoscope records from the 19[th] century and the
beginning of the 20[th] century in a new data collection. We include data from 23 observatories covering
Portugal and the African Atlantic regions, providing a latitudinal span from the extratropics in the
northern hemisphere to the tropics in the southern hemisphere. This record represents the most
extended ozonoscope data series recovered to date, spanning 50 years of daily data and 58 years of
monthly data, from 1855 to 1913. Moreover, with an small exception for part of a series of an
observatory, the existence of the observations here recovered had not been noticed in the previous
literature. This dataset presents only a small amount of inhomogeneities, and has the potential to
eventually bring unvaluable information on pre-industrial $O_3$. It exist plenty of data from other
observatories in logbooks that could be recovered (Bojkov, 1986; Möller, 2022), and such data and the
work here published can contribute to a better understanding of pre-industrial $O_3$ and serve for future
research on it.



**Author contribution**
J.A.A. devised the research, with the help of L.G. and A.S-L.; J.A.A. researched the books containing the datasets with the help of J.C.A.M., L.G., and M.A.V.; The datasets were digitised by J.A.A., J.C.A.M., and C.P.S.; Quality control on the data, including homogenization was performed by J.A.A., J.C.A.M, A.C.S., L.dlT. and Y.B. J.A.A., L.dlT. and L.G. secured the funding. J.A.A. and J.C.A.M. wrote the original draft. All authors have read and agreed to the published version of the manuscript.

**Competing interests**
The authors declare no competing interests regarding this paper.

**Acknowledgements**
We thank Stefan Brönnimann, Renate Varga (Auchmann) from the University of Bern, and Ricardo García-Herrera from the Universidad Complutense de Madrid for comments on early versions of this work and manuscript, especially to Ricardo García-Herrera for making us aware of the first meteorological records that kicked-off this research. Also, we would like to thank Amelie Driemel from PANGAEA for all the work related to the curation and publication of the datasets, and PANGAEA for storing and publishing them. This work was supported by the Government of Galicia under the project DOPPLER (Grant number: ED431F 2016/15). J.A.A. was supported by a grant of the "Programa de Recualificación del Profesorado Universitario" of the Spanish Ministry of Universities. C.P.S. is supported by the Government of Galicia under the grant ED481A-2024-246. A.C.S. thanks the action financed by the Ministry of Universities under application 33.50.460A.752 and by the European Union Next Generation EU/PRTR through a "*María Zambrano*" contract from the Universidade de Vigo, belonging to the launch of a European Recovery Instrument ("Next Generation EU"), aimed at requalifying the Spanish university system, specifically for teachers and attracting international talent. The EPhysLab is supported by the Government of Galicia (Grant number: ED431C 2021/44).

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





**Figures**
**Figure 1:** Map with the location of the observatories for which the data have been recovered.

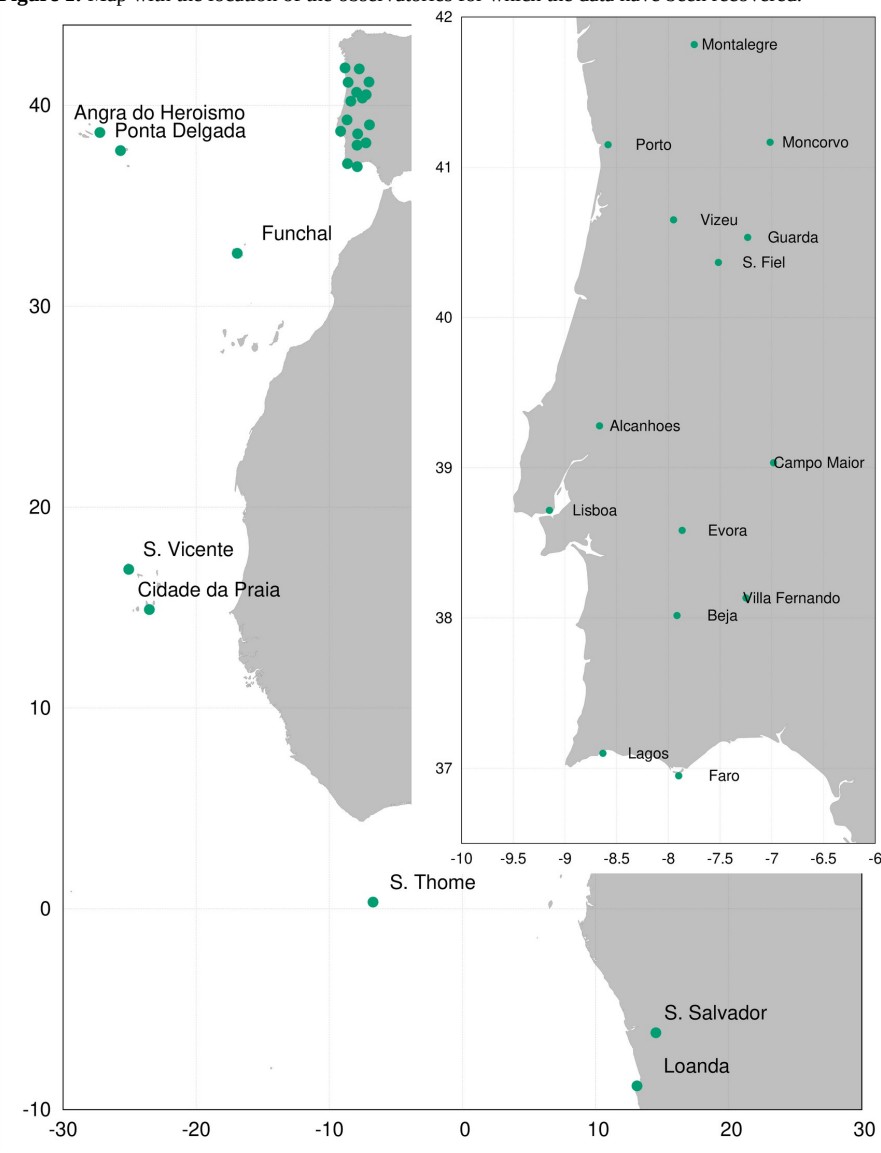




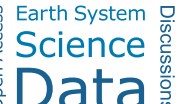

**Figure 2:** Example of the original tables from the AOIDL containing the recovered O₃ values on the
right hand column. Source: Brito Capello (1877).

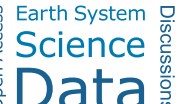




**Tables**
**Table 1**: List of observatories including country, region, latitude, longitude, and elevation. They are listed in decreasing latitude
order. Only monthly mean $O_3$ series are available from the four stations flagged with (*).

|  | Observatory | Country | Region | Lat. | Long. | E (m) |
|---|---|---|---|---|---|---|
| **1** | Montalegre | Portugal | Iberian Pen. | 41.82 | -7.75 | 1027 |
| **2** | Moncorvo | Portugal | Iberian Pen. | 41.17 | -7.02 | 415 |
| **3** | Porto | Portugal | Iberian Pen. | 41.15 | -8.58 | 100 |
| **4** | Vizeu | Portugal | Iberian Pen. | 40.65 | -7.95 | 494 |
| **5** | Guarda | Portugal | Iberian Pen. | 40.53 | -7.23 | 1039 |
| **6** | Serra da Estrela | Portugal | Iberian Pen. | 40.42 | -7.58 | 1450 |
| **7** | S. Fiel (*) | Portugal | Iberian Pen. | 40.37 | -7.52 | 516 |
| **8** | Alcanhoes (*) | Portugal | Iberian Pen. | 39.28 | -8.67 | — |
| **9** | Campo Maior | Portugal | Iberian Pen. | 39.03 | -6.98 | 288 |
| **10** | Infante D. Luiz | Portugal | Iberian Pen. | 38.72 | -9.23 | 95 |
| **11** | Angra do Heroismo | Portugal | Azores/ Macaronesia | 38.65 | -27.23 | 44 |
| **12** | Evora | Portugal | Iberian Pen. | 38.58 | -7.87 | 313 |
| **13** | Villa Fernando | Portugal | Iberian Pen. | 38.13 | -7.25 | 375 |
| **14** | Beja (*) | Portugal | Iberian Pen. | 38.02 | -7.92 | 284 |
| **15** | Ponta Delgada | Portugal | Azores/ Macaronesia | 37.75 | -25.68 | 20 |
| **16** | Lagos | Portugal | Iberian Pen. | 37.10 | -8.63 | 13 |
| **17** | Faro (*) | Portugal | Iberian Pen. | 36.95 | -7.90 | 14 |
| **18** | Funchal | Portugal | Madeira/ Macaronesia | 32.63 | -16.92 | 25 |
| **19** | S. Vicente | Cape Verde | Macaronesia | 16.90 | -25.07 | 11 |
| **20** | Cidade da Praia | Cape Verde | Macaronesia | 14.90 | -23.52 | 34 |
| **21** | S. Thome | Saint Thomas & Prince | Gulf of Guinea | 0.33 | -6.72 | 7 |
| **22** | S. Salvador do Congo | Angola | Continental Africa | -6.17 | 14.53 | 559 |
| **23** | Loanda | Angola | Continental Africa | -8.82 | 13.12 | 59 |





**Table 2:** Temporal coverage of the rescued daily O$_3$ for each of the 19 observatories (Infante D. Luiz and associated observatories). The stations are listed in decreasing order of the available number of years with data. Blue cells correspond to the years with data rescued. The numbers inside the cells represents the number of observations conducted daily. One daily observation (**1**) consisted in a 24-hour strip exposure, between 3 PM of consecutive days. Two daily observations (**2**) consisted in a 12-hour strip exposure from 9 AM to 9 PM and 9 PM until 9 AM of the following day. Blue cells without number indicate that no information was found about the number of daily observations.

| Year | Infante D. Luiz | Ponta Delgada | Funchal | Porto | Guarda | Campo Maior | Angra do Heroismo | Evora | São Thome | Moncorvo | Lagos | Montalegre | Vizeu | Loanda | Cidade da Praia | Serra da Estrela | Villa Fernando | São Vicente | São Salvador |
|---|---|---|---|---|---|---|---|---|---|---|---|---|---|---|---|---|---|---|---|
| 1913 | 2 | | | | | | | | | | | | | | | | | | |
| 1912 | 2 | | | | | | | | | | | | | | | | | | |
| 1911 | 2 | | | | | | | | | | | | | | | | | | |
| 1910 | 2 | | | | | | | | | | | | | | | | | | |
| 1909 | 2 | | | | | | | | | | | | | | | | | | |
| 1908 | 2 | | | | | | | | | | | | | | | | | | |
| 1907 | 2 | | | | | | | | | | | | | | | | | | |
| 1906 | 2 | | | | | | | | | | | | | | | | | | |
| 1905 | 2 | | | | | | | | | | | | | | | | | | |
| 1904 | 2 | | | | | | | | | | | | | | | | | | |
| 1903 | 2 | | | | | | | | | | | | | | | | | | |
| 1902 | 2 | | | | | | | | | | | | | | | | | | |
| 1901 | 2 | | | | | | | | | | | | | | | | | | |
| 1900 | 2 | | | | | | | | | | | | | | | | | | |
| 1899 | 2 | | | | | | | | | | | | | | | | | | |
| 1898 | 2 | | | | | | | | | | | | | | | | | | |
| 1897 | 2 | | | | | | | | | | | | | | | | | | |
| 1896 | 2 | | | | | | | | | | | | | | | | | | |
| 1895 | 2 | | | | | | | | | | | | | | | | | | |
| 1894 | 2 | | | | | | | | | | | | | | | | | | |
| 1893 | 2 | | | | | | | | | | | | | | | | | | |
| 1892 | 2 | | | | | | | | | | | | | | | | | | |
| 1891 | 2 | | | | | | | | | | | | | | | | | | |
| 1890 | 2 | | | | | | | | | | | | | | | | | | |
| 1889 | 2 | | | | | | | | | | | | | | | | | | |
| 1888 | 2 | 2 | 2 | | 1 | | 2 | 1 | 2 | 1 | | 1 | 2 | | | 2 | 2 | | |
| 1887 | 2 | 2 | 2 | 2 | 1 | 2 | 2 | 1 | 2 | 1 | | 1 | 2 | 2 | 2 | 2 | 2 | | |
| 1886 | 2 | 2 | 2 | 2 | 1 | 2 | 2 | 1 | 2 | 1 | | 1 | 2 | 2 | 2 | 2 | 2 | | |
| 1885 | 2 | 2 | 2 | 2 | 1 | 2 | 2 | 1 | 2 | 1 | | 1 | 2 | 2 | 2 | 2 | 2 | | |
| 1884 | 2 | 2 | 2 | 2 | 1 | 2 | 2 | 1 | 2 | 1 | | 1 | 2 | 2 | | 2 | 2 | | |
| 1883 | 2 | 2 | 2 | 2 | 1 | 2 | 2 | 1 | 1 | 1 | 1 | 1 | 2 | 2 | | 2 | 2 | | |
| 1882 | 2 | 2 | 2 | 2 | 1 | 2 | 2 | 1 | 1 | 1 | 1 | 1 | | 2 | 2 | | | | |
| 1881 | 2 | 2 | 2 | 2 | 1 | 2 | 2 | 1 | 1 | 1 | 1 | 1 | 2 | 2 | | | | | |
| 1880 | 2 | 2 | 2 | 2 | 1 | 2 | 2 | 1 | 1 | 1 | 1 | 1 | | | | | | | |
| 1879 | 2 | 2 | 2 | 2 | 2 | 2 | 2 | 1 | 1 | 1 | 1 | 1 | | | | | | | |
| 1878 | 2 | 2 | 2 | 2 | 2 | 2 | 2 | 1 | 1 | 1 | 1 | 1 | | | | | | | |
| 1877 | 2 | 2 | 2 | 2 | 1 | 2 | 2 | 1 | 1 | | 1 | | | | | | | | |
| 1876 | 2 | 2 | 2 | 2 | 1 | 2 | 2 | 1 | 1 | | 1 | | | | | | | | |
| 1875 | 2 | 2 | 2 | 2 | 1 | 2 | 2 | 1 | 1 | | 1 | | | | | | | | |
| 1874 | 2 | 2 | 2 | 2 | 1 | 2 | 2 | 1 | 1 | | 1 | | | | | | | | |
| 1873 | 2 | 2 | 2 | 2 | | 2 | 2 | 1 | | | | | | | | | | | |
| 1872 | 2 | 2 | 2 | 2 | 1 | 2 | 2 | 1 | | | | | | | | | | | |
| 1871 | 2 | | | | | | | | | | | | | | | | | | |
| 1870 | 2 | | | | | | | | | | | | | | | | | | |
| 1869 | 2 | | | | | | | | | | | | | | | | | | |
| 1868 | 2 | | | | | | | | | | | | | | | | | | |
| 1867 | 2 | | | | | | | | | | | | | | | | | | |
| 1866 | 2 | | | | | | | | | | | | | | | | | | |
| 1865 | 2 | | | | | | | | | | | | | | | | | | |
| 1864 | 2 | | | | | | | | | | | | | | | | | | |
| 1863 | 2 | | | | | | | | | | | | | | | | | | |



**Table 3:** Temporal coverage of the rescued monthly mean $O_3$ for each of the observatories: period rescued and available data. Brown cells correspond to the years with data rescued. The numbers inside the cells represent the number of monthly means available per year.

| Station | 1855 | 1856 | 1857 | 1858 | 1859 | 1860 | 1861 | 1862 | 1863 | 1864 | 1865 | 1866 | 1867 | 1868 | 1869 | 1870 | 1871 | 1872 | 1873 | 1874 |
|---|---|---|---|---|---|---|---|---|---|---|---|---|---|---|---|---|---|---|---|---|
| Alcanhoes | | | | | | | | | | | | | | | 9 | 12 | 3 | | | |
| Angra do Heroismo | | | | | | | | | | | | 3 | 11 | 12 | 12 | 11 | 12 | 11 | | |
| Beja | | | | | | | | | | | | | | | | | | | | |
| Campo Maior | | | | | | | | | 1 | 12 | 12 | 12 | 12 | 12 | 12 | 12 | 12 | 11 | | |
| Cidade da Praia | | | | | | | | | | 1 | 12 | 5 | | | | | | | | |
| Evora | | | | | | | | | | | | | | | 1 | 12 | 12 | 12 | 12 | 12 |
| Faro | | | | | | | | | | | | | | | | | | | | |
| Funchal | | | | | | | | | | 1 | 11 | 11 | 12 | 12 | 12 | 12 | 12 | 11 | 12 | 12 |
| Guarda | | | | | | | | | 12 | 12 | 12 | 12 | 12 | 12 | 12 | 12 | 12 | 12 | 12 | 12 |
| Illa de S. Vicente | | | | | | | | | | | | | | | | | | | | |
| Infante D Luiz | 1 | 12 | 12 | 12 | 12 | 12 | 12 | 12 | 12 | 12 | 12 | 12 | 12 | 12 | 12 | 12 | 12 | 12 | 12 | 12 |
| Lagos | | | | | | | | | | | | | | | | | | 6 | 6 | 10 |
| Loanda | | | | | | | | | | | | | | | | | | | | |
| Moncorvo | | | | | | | | | | | | | | | | | | | | |
| Montalegre | | | | | | | | | | | | | | | | | | | | |
| Ponta Delgada | | | | | | | | | | | 10 | 12 | 12 | 12 | 11 | 10 | 12 | 12 | 11 | 12 |
| Porto | | | | | | | | 1 | 12 | 12 | 12 | 12 | 12 | 11 | 12 | 12 | 12 | 12 | 12 | 12 |
| S. Fiel | | | | | | | | | | | | | | | | | | | | |
| S. Thome | | | | | | | | | | | | | | | | | | 5 | 8 | 12 |
| Salvador do Congo | | | | | | | | | | | | | | | | | | | | |
| Serra da Estrela | | | | | | | | | | | | | | | | | | | | |
| Villa Fernando | | | | | | | | | | | | | | | | | | | | |
| Vizeu | | | | | | | | | | | | | | | | | | | | |

| Station | 1875 | 1876 | 1877 | 1878 | 1879 | 1880 | 1881 | 1882 | 1883 | 1884 | 1885 | 1886 | 1887 | 1888 | 1889 | 1890 | 1891 | 1892 | 1893 | 1894 |
|---|---|---|---|---|---|---|---|---|---|---|---|---|---|---|---|---|---|---|---|---|
| Alcanhoes | | | | | | | | | | | | | | | | | | | | |
| Angra do Heroismo | | | | | | | | | | | | | | 12 | 12 | 12 | 12 | 12 | 12 | 12 |
| Beja | | | | | | | | | | | | | | | | | | | | |
| Campo Maior | | | | | | | | | | | | | | 10 | 11 | 12 | 12 | 12 | 12 | |
| Cidade da Praia | | | | | | | | | | | | | | | 11 | 12 | | | | |
| Evora | 12 | 12 | 12 | 12 | 12 | 12 | 12 | 12 | 12 | 12 | 12 | 12 | 12 | 12 | 12 | 12 | 12 | 12 | 12 | 12 |
| Faro | | | | | | | | | | | | | | | | | | | | |
| Funchal | 12 | 11 | 12 | 12 | 10 | 12 | 12 | 10 | 12 | 10 | 11 | 11 | 12 | 12 | 12 | 12 | 11 | 12 | | 11 |
| Guarda | 12 | 11 | 12 | 12 | 12 | 12 | 11 | 12 | 12 | 12 | 12 | 12 | 12 | 12 | 12 | 12 | 12 | 12 | 12 | 12 |
| Illa de S. Vicente | | | | | | | | | | 12 | 11 | 10 | 10 | 11 | 12 | 12 | | 12 | 12 | 11 |
| Infante D Luiz | 12 | 12 | 12 | 12 | 12 | 12 | 12 | 12 | 12 | 12 | 12 | 12 | 12 | 12 | 12 | 12 | 12 | 12 | 12 | 12 |
| Lagos | 12 | 12 | 11 | 12 | 11 | 12 | 12 | 5 | | | | | | | | | | | | |
| Loanda | | | | | | 12 | 12 | 12 | 12 | 12 | 12 | 12 | 12 | | | | | | | |
| Moncorvo | | | 12 | 12 | 12 | 12 | 12 | 12 | 12 | 12 | 12 | 12 | 12 | 12 | 12 | 12 | 12 | 12 | 12 | 12 |
| Montalegre | | | | | 12 | 12 | 12 | 12 | 12 | 12 | 12 | 12 | 12 | 12 | 12 | 12 | 12 | 12 | 12 | 12 |
| Ponta Delgada | 12 | 12 | 12 | 12 | 12 | 12 | 12 | 12 | 12 | 12 | 12 | 12 | 12 | 12 | 12 | 12 | 12 | 12 | 12 | 12 |
| Porto | 12 | 11 | 12 | 12 | 12 | 12 | 12 | 12 | 12 | 12 | 11 | 12 | 12 | | | | | | | |
| S. Fiel | | | | | | | | | | | | | | | | | | | | |
| S. Thome | 12 | 8 | 12 | 12 | 12 | 12 | 12 | 10 | 10 | 6 | | 12 | 12 | 12 | 12 | | | | | |
| Salvador do Congo | | | | | | | | | 6 | 4 | 11 | 11 | 12 | 8 | | | | | | |
| Serra da Estrela | | | | | | | | 11 | 12 | 12 | 12 | 11 | 11 | 12 | 12 | 12 | 11 | 10 | 11 | 12 |
| Villa Fernando | | | | | | | | | | 7 | 12 | 12 | 12 | 12 | 12 | 12 | 12 | 12 | 12 | 12 |
| Vizeu | | | | | 12 | 12 | 12 | 9 | 12 | 12 | 12 | 12 | 12 | | | | | | | |

| Station | 1895 | 1896 | 1897 | 1898 | 1899 | 1900 | 1901 | 1902 | 1903 | 1904 | 1905 | 1906 | 1907 | 1908 | 1909 | 1910 | 1911 | 1912 | 1913 |
|---|---|---|---|---|---|---|---|---|---|---|---|---|---|---|---|---|---|---|---|
| Alcanhoes | | | | | | | | | | | | | | | | | | | |
| Angra do Heroismo | 12 | 12 | 12 | 12 | 12 | 12 | 12 | 12 | 12 | | | | | | | | | | |
| Beja | | | 12 | 12 | 12 | 12 | 12 | 12 | 12 | 12 | 12 | | | | | | | | |
| Campo Maior | | | | | | | | | | | | | | | | | | | |
| Cidade da Praia | | | | | | | | | | 12 | 12 | | | | | | | | |
| Evora | 12 | 12 | 12 | 12 | 12 | 12 | | | 12 | 12 | | | | | | | | | |
| Faro | 12 | 12 | 12 | 12 | 12 | 11 | 11 | 11 | 12 | 12 | 12 | | | | | | | | |
| Funchal | | 12 | 12 | 12 | 11 | 11 | 12 | 12 | 12 | 12 | 12 | | | | | | | | |
| Guarda | 12 | 12 | 12 | 11 | 12 | 12 | 12 | 12 | 12 | 12 | 12 | | | | | | | | |
| Illa de S. Vicente | 12 | 12 | 12 | 12 | 12 | 12 | | | | | | | | | | | | | |
| Infante D Luiz | 12 | 12 | 12 | 12 | 12 | 12 | 12 | 12 | 12 | 12 | 12 | 12 | 12 | 12 | 12 | 12 | 12 | 12 | 10 |
| Lagos | | | | | | | | | | 11 | 12 | | | | | | | | |
| Loanda | | | | | | | | | | | | | | | | | | | |
| Moncorvo | 12 | 12 | 12 | 12 | 12 | 12 | 12 | 12 | 12 | 12 | 12 | | | | | | | | |
| Montalegre | 12 | 12 | 12 | 12 | 12 | 12 | 12 | 12 | 12 | 12 | 12 | | | | | | | | |
| Ponta Delgada | 12 | 12 | 12 | 12 | 12 | 12 | 12 | 12 | 12 | | | | | | | | | | |
| Porto | | | 12 | | | 12 | 12 | | | | | | | | | | | | |
| S. Fiel | | | | | | | | 11 | 12 | 12 | 12 | | | | | | | | |
| S. Thome | | | | | | | | | | | | | | | | | | | |
| Salvador do Congo | | | | | | | | | | | | | | | | | | | |
| Serra da Estrela | 12 | 12 | 12 | 10 | 9 | 12 | 12 | 12 | 11 | 12 | 12 | | | | | | | | |
| Villa Fernando | 12 | 12 | 12 | 12 | 12 | 12 | 11 | 11 | 12 | 12 | 12 | | | | | | | | |
| Vizeu | | | | | | | | | | | | | | | | | | | |





**Table 4:** Observatory, date and SNHT value for the $O_3$ breakpoints.

| Observatory | Date | SNHT |
|---|---|---|
| Angra do Heroismo | 1891-10-01 | 59.8 |
| Beja | 1902-11-01 | 31.2 |
| Campo Maior | 1866-05-01 | 25.8 |
| Campo Maior | 1888-01-01 | 27.6 |
| Campo Maior | 1891-12-01 | 39.2 |
| Evora | 1874-01-01 | 28.4 |
| Evora | 1890-07-01 | 66.7 |
| Evora | 1899-11-01 | 45.4 |
| Funchal | 1870-09-01 | 28.2 |
| Funchal | 1879-11-01 | 25.7 |
| Funchal | 1885-04-01 | 48.2 |
| Funchal | 1901-01-01 | 33.4 |
| Guarda | 1864-01-01 | 73.9 |
| Guarda | 1867-11-01 | 25.6 |
| Guarda | 1871-08-01 | 35.8 |
| Guarda | 1885-05-01 | 55.5 |
| Guarda | 1887-10-01 | 26.7 |
| Guarda | 1896-07-01 | 55.7 |
| Infante D. Luiz | 1866-06-01 | 30.9 |
| Infante D. Luiz | 1875-10-01 | 30.6 |
| Infante D. Luiz | 1879-04-01 | 70.6 |
| Infante D. Luiz | 1883-05-01 | 54.4 |
| Infante D. Luiz | 1889-12-01 | 27.0 |
| Lagos | 1904-02-01 | 65.2 |
| Moncorvo | 1878-11-01 | 36.8 |
| Moncorvo | 1879-08-01 | 43.9 |
| Moncorvo | 1887-09-01 | 87.0 |
| Moncorvo | 1904-10-01 | 26.9 |
| Montalegre | 1880-06-01 | 27.8 |
| Montalegre | 1892-05-01 | 69.9 |
| Ponta Delgada | 1867-12-01 | 88.6 |
| Ponta Delgada | 1898-09-01 | 40.1 |
| Ponta Delgada | 1902-06-01 | 25.1 |
| Porto | 1863-12-01 | 91.6 |
| Porto | 1886-10-01 | 37.5 |
| Porto | 1900-06-01 | 30.3 |
| Porto | 1900-12-01 | 26.4 |
| S. Thome | 1874-10-01 | 28.7 |
| S. Thome | 1886-01-01 | 76.6 |
| S. Vicente | 1886-07-01 | 36.9 |
| S. Vicente | 1892-02-01 | 32.2 |
| S. Vicente | 1895-04-01 | 36.8 |
| Serra da Estrela | 1889-08-01 | 26.6 |
| Villa Fernando | 1890-02-01 | 62.8 |
| Villa Fernando | 1896-07-01 | 75.2 |
| Villa Fernando | 1903-02-01 | 30.8 |
| Vizeu | 1882-10-01 | 31.6 |






**Table 5:** Observatory, date and SNHT value for the RH breakpoints.

| Observatory | Date | SNHT |
|---|---|---|
| Angra do Heroismo | 1888-05-01 | 55.1 |
| Angra do Heroismo | 1891-10-01 | 53.0 |
| Angra do Heroismo | 1902-05-01 | 15.7 |
| Campo Maior | 1864-07-01 | 16.5 |
| Campo Maior | 1872-11-01 | 31.5 |
| Campo Maior | 1890-06-01 | 17.4 |
| Cidade da Praia | 1904-06-01 | 21.7 |
| Evora | 1873-10-01 | 28.2 |
| Evora | 1878-04-01 | 37.0 |
| Evora | 1883-10-01 | 15.5 |
| Evora | 1888-12-01 | 39.5 |
| Evora | 1904-03-01 | 15.1 |
| Faro | 1904-04-01 | 15.6 |
| Funchal | 1871-10-01 | 24.8 |
| Funchal | 1884-06-01 | 16.6 |
| Funchal | 1888-11-01 | 32.1 |
| Funchal | 1894-03-01 | 39.3 |
| Funchal | 1896-11-01 | 18.6 |
| Guarda | 1879-08-01 | 69.1 |
| Guarda | 1887-09-01 | 20.3 |
| Guarda | 1895-10-01 | 16.2 |
| Guarda | 1904-04-01 | 17.8 |
| Infante D Luiz | 1863-10-01 | 28.4 |
| Infante D Luiz | 1866-09-01 | 15.3 |
| Infante D Luiz | 1873-12-01 | 18.1 |
| Infante D Luiz | 1891-09-01 | 21.9 |
| Loanda | 1881-05-01 | 19.4 |
| Loanda | 1884-10-01 | 17.4 |
| Moncorvo | 1879-10-01 | 27.9 |
| Moncorvo | 1887-06-01 | 59.1 |
| Montalegre | 1894-11-01 | 15.2 |
| Ponta Delgada | 1867-12-01 | 65.1 |
| Ponta Delgada | 1887-04-01 | 19.3 |
| Ponta Delgada | 1894-03-01 | 28.3 |
| Ponta Delgada | 1896-08-01 | 18.9 |
| Porto | 1882-03-01 | 27.4 |
| Porto | 1883-09-01 | 17.1 |
| Porto | 1885-01-01 | 63.9 |
| Porto | 1885-08-01 | 15.9 |
| Porto | 1887-06-01 | 78.3 |
| S. Fiel | 1902-07-01 | 16.0 |
| S. Fiel | 1904-10-01 | 18.1 |
| S. Salvador do Congo | 1885-11-01 | 21.1 |
| S. Vicente | 1889-03-01 | 24.2 |
| S. Vicente | 1890-12-01 | 30.2 |
| Vizeu | 1882-04-01 | 16.8 |
