# Peer review of "19th–20th century semi-quantitative surface ozone along 1 subtropical Europe to tropical Africa Atlantic coasts 2 3"

_Earth System Science Data, 2024_

## Author Comment (AC1)

https://doi.org/10.5194/essd-2024-366-RC1

**RC1**: Anonymous Referee #1, 20 Dec 2024

The paper is well-written and the dataset will potentially be of interest to at least three groups of readers: climate research, ozone research, and science historiography.

For climate research, the dataset presents the interesting aspect of adding several decades of station data that predate what is taken as the acceptable observational record for surface ozone. The authors review the previously-published datasets and present appropriately the originality of the new dataset. The authors warn that exploitation will not be straightforward, for reasons cited in the paper, chiefly a strong dependence of the semi-quantitative ozone obtained by ozonoscope on humidity and other factors. The papers having reviewed the data quality from these instruments are also duly cited. However, to assit data exploitation, it seems humidity data are also available along with the observations. Furthermore, as data from other neighboring observatories and stations are rescued -in a similar way as done here- the accuracy of reconstruction of the humidity information is due to improve with techniques such as climate reanalysis (e.g., Slivinski et al., 2019, https://doi.org/10.1002/qj.3598).

One point to clarify would be if all observed elements reported in the original sheets got transcribed? or if some observed elements were not digitized, which ones were left out?

The reviewer makes a good point here. We understand that the reviewer is asking if in the logbooks from observatories there was more information on other variables measured. The answer for this is yes. This is shown in our Fig.2, which on top of the ozone and humidity data, it includes pressure, temperature, winds, etc. Winds were digitized for Lisbon, but for different purposes to the goals of this paper and irrelevant at the moment for the purpose of this research. Therefore, they are not included in the published database. and other variables were not digitized. We clarify this now in the text, modifying the sentence at the beginning of Section 2.2, which now reads "*usually consisting of reports of atmospheric pressure, rain, evaporation, temperature, vapor pressure, humidity, cloud coverage, wind direction and speed. Also, they included $O_3$, published for the first time in 1863, beginning the series of "Annaes do Observatorio do Infante D. Luiz" (hereinafter AOIDL) reports (see Figure 2 for an example) (Brito Capello, 1863). Only $O_3$ and humidity data were digitized in the context of the work presented here.*".

For ozone research, the dataset presents, besides monthly data, also daily nightly and diurnal observations (when available). This availability within the data series may present an opportunity to apply to earlier times our understanding of ozone chemistry and surface ozone variability, which is still developing (e.g., Monks et al., 2021, https://doi.org/10.5194/acp-21-12909-2021).

Many thanks for bringing this on the table. It is an obvious opportunity that these datasets enable, and we had not thought about suggesting it. We have adapted the comment of the reviewer to include it in the Conclusions section of our manuscript, and now we cite the paper by Monks et al.

Finally, for science historiography, this paper presents the interesting point of illustrating how scientists today recognize, as a community, when observation techniques do evolve, and thus for some applications it may be preferable to leave aside some of the earlier records (e.g., Tarasick et al., 2019), but even then, how efforts continue to try and extract value from past observations (this paper).

We fully agree with this reflection by the reviewer, and we are happy that this view is shared by others in the research community. We hope that this is a useful paper and contributes to this idea and the discussion on it.

Table 3: the image would need to be rotated (years are shown upside-down).

We are aware of it. However, this is a problem of the format of the manuscript to submit to the journal. Both Table 2 and Table 3 should be in landscape, instead of vertical, as they appear right now. When the tables are in landscape, the numbers in the vertical are not flipped and read well. We will work with the typewriters to get it right before publication of the final manuscript.

---

## Author Comment (AC2)

**https://doi.org/10.5194/essd-2024-366-RC2**

**RC2**: Anonymous Referee #2, 08 Jan 20253

This paper presents a new dataset of semi-quantitative ozone concentrations in several locations of Iberia and western Africa for early periods. Despite ozone measurements make with test-paper has been regarded as "not recommended for quantitative use" because of the sensitivity of test-papers to relative humidity, I consider that the data presented are very interesting and they deserve publication. Besides, the authors present a complete discussion about the characteristics of the data and its limitations, so future users are aware of the problems.

The paper is clear, concise and well written. The presented data are accessible and can be useful for future users, so I recommend publication. I only have some minor comments, mostly formal, that in my opinion could help to improve the clarity of the paper and the interpretation of the data.

Line 74. Consider adding a reference to Figure 1 here.

Thanks, We have added it now.

Lines 109-110. Here it is established that the ozone concentration was originally quantified in a scale from 1 to 11 and then (line 118), it is said that this scale was later upgraded to one from 0 to 21. It is hard to know what system is used in the files and this is relevant for future users. Maybe labeling "[O3] Schönbein scale" or "[O3] Berigny scale" instead of "[O3] arbitrary units" in the data files would clarify the issue.

It is not that the original scale was upgraded, but that many other scientists and manufacturers of meteorological instrumentation made their own adaptations of the original Schönbein paper and scale. Even there were punctual scales going between 0 and 14 or 0 and 16. Bojkov (1986) provides information on it. However, by far, the two more popular were the ones by Schönbein and Bérigny. Actually, in the reports that we present here, the original values were measured in the Bérigny scale, and then divided by two, to make them closer to the Schönbein scale, we guess that trying to make them comparable to previous existing records in such scale. We have added a clarification with all this information in the "Comments" section of the data repositories in PANGAEA, and the headers of the published files, which now read: "*The semi-quantitative method for the ozone measurement is unitless. There were 2 main chromatic scales for the same range of color from the original color of the paper to different tonalities of purple. This range of tonalities (function of the Ozone concentration) was 0 to 10 in the Schönbein scale (Schönbein, 1850) and 0 to 21 in the Bérigny scale (Bérigny, 1858). In the case of the files reported here, the original measurements were done in the Bérigny scale; however, to annotate them in the logbooks, the original Bérigny values were divided by two to make them someway coincident with the Schönbein scale as mentioned in the Infante Dom Luiz logbooks (e.g. Fradesso da Silveira, 1865, pages 13 and 63).*"

In addition, in the paper it is said that the Schönbein scale goes from 1 to 11 (line 109) and the Berigny scale goes from 0 to 21 (line 118). However in the files (at least in the ones I tested) it is said "This range of tonalities (function of the Ozone concentration) was 1 to 10 in the Schönbein scale and 1 to 21 in the Berigny scale". Please clarify the discrepancies.

Thanks to the reviewer for detecting these incongruities. Actually, the issues with the

scales, as pointed in the previous comment, were more complex. We have now corrected them both in the manuscript and the repository, clarifying the issues with the different scales, and making the information more homogeneous.

Paragraph from line 145 to 158. Consider adding a reference to Figure 2 for clarity.

We include now a reference to Figure 2.

Line 240. Do the authors mean tables 4 and 5?

Yes, the reviewer is right. Many thanks for pointing it out. This mistake was caused by a correction requested by the editorial office of the journal before posting the manuscript in Discussions and after submitting it. The office requested to label the mentioned two tables as such instead of figures; that is what they are in the document. Therefore, this semantic issue made the mention in this line go unnoticed and wrong. We will work with the copy-editing service of the journal to get it right, as figures or tables, if the manuscript is accepted for publication.

Table 3. Please add grid lines for clarity.

Done.